# Key stakeholders' perspectives and experiences with defining, identifying and displaying gaps in health research: a qualitative study protocol

Linda Nyanchoka,[1,2,3] Catrin Tudur-Smith,[2,3] Raphaël Porcher,[1] Darko Hren[4]

¹Université de Paris, CRESS, INSERM, INRA, F-75004, Paris, France
²Institute of Translational Medicine, University of Liverpool, Liverpool, UK
³Department of Biostatistics, University of Liverpool, Liverpool, UK
⁴School of Humanities and Social Sciences, University of Split, Split, Croatia

**Correspondence to**
Linda Nyanchoka;
lnyanchoka@gmail.com

## ABSTRACT

**Introduction** Identifying research gaps can inform the design and conduct of health research, practice and policies by informing the current body of evidence. Audiences including researchers, clinical guideline developers, clinicians, policymakers, research regulatory bodies, funders and patients/the public can also benefit from understanding the status of research and research gaps to make informed choices. This study aims to explore how key informants define research gaps and characterise methods/practices used to identify and display gaps in health research to inform future research practice and policies.

**Methods and analysis** This is an exploratory qualitative study using semi-structured in-depth interviews. The participants will be recruited by purposive sampling from initiatives and organisations previously identified in a scoping review on methods to identify, prioritise and display gaps in health research. We anticipate performing up to 28 interviews with the different key informant groups who are involved in using evidence to inform health policy, practice and research. Interviews will be thematically analysed as outlined by Braun and Clarke. The qualitative data-analysis software NVivo V.12 Pro will be used to aid data management and analysis.

**Discussion** This is the protocol for a follow-up study that aims to complement and enrich the findings of the scoping review on methods to identify, prioritise and display gaps in health research. The overall project aims to develop methodological guidance for describing, identifying and displaying gaps in health research.

**Ethics and dissemination** The research obtained ethical approval from the University of Liverpool, UK. The findings will be disseminated via conferences, meetings (organised by the Methods in Research on Research project), peer-reviewed publications and lay magazines because the study participants will include the public/patients.

## Strengths and limitations of this study

► The qualitative nature of this study provides an in-depth understanding of key informants' perspectives and experiences in describing, identifying and displaying gaps in health research.
► This study is embedded in a larger study aiming to develop methodological guidance to identify and display gaps in health research.
► This study would have benefited from including patient/public perspectives in designing the study to be able to improve the importance and relevance of the findings for this population.

scoping review on methods used to identify, prioritise and display gaps in health research reported 12 different definitions related to gaps in health research (eg, population, theoretical and methodology gaps), each describing research gaps differently.[1] This finding shows the ambiguity of the term 'research gaps' and the different practices it may be related to.

As a basis for further exploring and understanding 'research gaps', we start from the definition given by the National Collaborating Centre for Methods and Tools (NCCMT) in Canada based on the work of Robinson *et al*, whereby a research gap is defined as a topic or area for which missing or insufficient information limits the ability to reach a conclusion for a question.[2] Given the different meanings and definitions of research gaps found in the scoping review,[1] we consider it important to further explore definitions rather than just adopt or modify the NCCMT definition. Clearly defining the type of research gap can help determine how to better identify, characterise, prioritise and address research gaps.

Different methods for identifying research gaps have been reported; for example, scoping reviews and umbrella reviews are emerging methods for mapping and

## BACKGROUND

Identifying research gaps can help inform the design and conduct of health research, practice and policies by providing a better understanding of the current body of evidence. The term 'research gap' is not well defined, and its meaning can differ depending on the researcher and research context. A recent

summarising evidence. These methods have an explicit aim of identifying research gaps in a broad area as compared with systematic reviews that focus on answering a specific research question.[3–7] Robinson *et al* developed a framework using systematic reviews to identify research gaps[2] in which they classified the reasons for the existence of research gaps and used the population, intervention, comparison, outcome and setting process to characterise them. Scoping, umbrella and systematic reviews are reported to specifically identify research gaps, but other methods are being used, and further exploring these methods can optimise their definition, methodological scrutiny and practice.[8–18] Furthermore, the aforementioned methods focus on the use of secondary research methods to identify research gaps. However, a recent scoping review showed that other methods have been used to identify gaps, including primary and both primary and secondary research methods.[1] The scoping review showed a lack of consensus on what constitutes the best methodological approaches to identify research gaps, determine research priorities and display research gaps or priorities.[1 5 7] Therefore, to better understand the different methods and ongoing practices, we aimed to conduct a qualitative study to further explore more in-depth key stakeholder experiences in describing research gaps and the methods used to identify and display gaps in health research.

This study is part of larger ongoing efforts to avoid waste in producing and reporting research evidence, with a focus on the identification of research gaps.[19] Healthcare decisions for individual patients, public health policies and clinical guidelines should be informed by the best available research evidence while taking into consideration research gaps. Investigating experiences with practices/methods used to identify research gaps can inform explicit methodological approaches in identifying and describing research gaps. This investigation can enhance practices of different stakeholder groups (ie, health professionals, commissioners, researchers, patients/the public and decision-makers) when addressing areas of uncertainty within the research problem and topic area.[20] Initiatives such as the James Lind Alliance, UK Database of Uncertainties about the Effects of Treatments, Cochrane Agenda and Priority Setting Methods Group and Evidence-based Research Network are some examples of existing efforts to identify and prioritise research gaps in health.[1]

This study is nested in a larger project aimed at developing methodological guidance for identifying gaps in health research. The first step in the project was a scoping review describing methods used to identify, prioritise and display gaps in health research in scientific literature. The scoping review mapped evidence on different definitions reported for the term 'research gap' as well as methods used to identify research gaps and determine research priorities and display research gaps or research priorities.[1] The second step is the qualitative study described in this protocol. The aim of the study is to investigate the

experience of key stakeholders (ie, researchers, funders, clinicians, clinical guideline developers, public health professionals, commissioners, patients/the public and policymakers) with defining research gaps and practices/methods used to identify and display research gaps. The final step will be an integration and overview combining findings from the scoping review and qualitative study to provide a comprehensive overview of methods used to identify and display research gaps. These study findings will be used to inform the methodological guidance on identifying research gaps.

The specific objectives of the study are to (1) investigate key stakeholders' knowledge, perceptions and experiences with defining research gaps and (2) characterise methods/practices used for identifying and displaying gaps in health research.

## METHODS AND ANALYSIS
### Qualitative study design
This study is an exploratory qualitative study using semi-structured interviews. This method will provide in-depth insight into key stakeholders' perspectives, experiences, and practices with defining, identifying and displaying research gaps. Investigating perspectives of different key stakeholders will ensure that the issue is not explored through one lens but rather a variety of lenses. This will allow for revealing and better understanding multiple facets of research gaps including definitions and methodological approaches/practices to identify and display gaps.[21]

### Study sample and recruitment
The study sample will include the following stakeholder groups (ie, researchers, funders, clinicians, clinical guideline developers, public health professionals, commissioners, patients/the public and policymakers). The stakeholder groups will be organised in three main categories focusing on the use of evidence to inform health policy, health practice and health research. These categories (policy, practice and research) are determined from the scoping review findings.[1] More information and examples of organisations are given in table 1. Study participants will be recruited via contacts and organisations identified in the scoping review, relevant scientific publications, existing professional networks (eg, Horizon 2020 (H2020) Project Methods in Research on Research (MiRoR)) and contacts from conference attendance (eg, Evidence Live and Cochrane Colloquium).

This study will also include patients or members of the public as key informants, which will allow for better understanding participants' perceived needs and priorities in identifying research gaps to make informed health decisions. Patients/the public will be recruited and identified via patient support groups online, community centres and public involvement websites such as the peopleinresearch.org platform that involves the public in health research.

**Table 1** Key informants

| Categories | Key informants | Examples | Expected number of interviews |
|---|---|---|---|
| Health policy | Policymakers | Ministry of health officials | 2–4 |
| Health practice | Clinicians | Healthcare professionals (doctors, nurses) | 2–4 |
| | Clinical guideline developers | UK National Institute for Health and Care Excellence | 2–4 |
| | Public health professionals, commissioners | National public health bodies | 2–4 |
| | Public/patients | Patient forums/groups | 2–4 |
| Health research | Researchers | Research institutes/universities Knowledge synthesis research groups Belgian Health Care Knowledge Centre Africa Evidence Network Student forums | 2–4 |
| | Funding bodies | UK National Institute for Health Research European Union | 2–4 |

We will use purposive sampling to ensure that the perspectives of all identified stakeholder groups are represented. Purposeful sampling is widely used in qualitative research for identifying and selecting information-rich cases, and in this study, further elaboration of the term research gap is needed to better understand the context of the research gaps and methods/practices used to identify and display the research gaps.[22 23]

We anticipate performing about 14–28 interviews. This number of interviews will provide for data saturation (ie, the point when new data do not add to a better understanding of the studied phenomenon but rather repeat what was previously expressed[24]) and also obtain a scope of responses from each stakeholder group. This estimation of interview participants is based on a study involving 60 interviews that showed saturation with 12 interviews, with broader themes apparent after only 6 interviews.[25] The authors noted that factors such as heterogeneity of the sample affect how many interviews are required but concluded that to understand common perceptions and experiences among a group of relatively homogeneous individuals, 12 interviews should suffice.[25] Another study, after examining 25 in-depth interviews, found code saturation after interviews, with the range of thematic issues identified; the authors proposed 16–24 interviews to reach saturation (ie, a richly textured understanding of issues[26]). Therefore, we aim to gather 14–28 interviews for our three main categories (health policy, practice and research).

Saturation will be guided by the seven parameters identified by Hennink et al,[26 27] including the study purpose, population, sampling strategy, data quality, type of codes, code book and saturation goal, and focus retrieved from the study. Each of these parameters will be considered throughout the study.

## Data collection and recording

Semi-structured interviews will be used for this study. The main reason for selecting semi-structured interviews is to allow for specific areas to be addressed while giving the interviewees the opportunity to reflect on their experiences and perspectives related to defining, identifying and presenting research gaps that are relevant to them and that may not have been explored or anticipated by the researcher(s).[28]

We will conduct interviews in-person and using teleconference, according to the participant's availability and preference. In-person interviews will be conducted primarily with participants residing or reachable in London, UK, and other participants will be interviewed via teleconference (see online supplementary appendix 1 for the interview guide for both the in-person and teleconference interviews). The interviews will be recorded on a digital recorder for face-to-face interviews and electronically for teleconference interviews.

The guide was developed by focusing on exploring key stakeholder perspectives and experiences with the following key areas:
1. Participant background information.
2. Definitions of research gaps.
3. Knowledge, perceptions and experiences on methods/practices used to identify and display gaps in health research to inform further health policy, practice and research.

These three domains were developed with information from the scoping review to guide the questions. The interview topic guide will be piloted before data collection. It will also be adapted according to key stakeholder groups to ensure that it is meaningful to their background and to gather more relevant information based on their experiences and knowledge.[29]

The semi-structured interview guide contains two levels of questions: main themes and follow-up questions. The main themes cover the general content of the research gaps aimed to encourage participants to speak freely about their perceptions, experiences and practices. Follow-up questions are prompts and probes aimed at following respondents' answers and investigating the raised issues more in-depth. The interview guide covers the main topics of the study, providing a focused structure for the discussion during the interviews. However, it does not need to be strictly followed — the main focus is on providing a setting that encourages respondents to share their perceptions and experiences with research gaps as thoroughly as possible within the constraints of our study aims.[30]

All interviews will be transcribed verbatim and anonymised. The lead researcher (LN) will transcribe two interviews to help inform the analytical process, and the other audio files will be transcribed by a professional transcription agency licensed from the University of Liverpool.

## Data analysis

We will use analytical categories to describe and explain definitions, experiences and practices reported among the groups of participants. All data relevant to each category (describing research gaps, experience with identifying and displaying research gaps) will be identified and examined to ensure that each data item is checked accordingly.

Our approach is based on the thematic analysis outlined by Braun and Clarke.[31] The steps include the following: (1) transcription and checking transcripts with recordings for accuracy; (2) open coding from interview responses to be performed by two researchers independently (LN and DH); (3) agreement of initial codes to be discussed among the researchers and an initial codebook developed; (4) the code structure to be used for analysing the remaining responses with openness to including new codes and refining existing ones; and (5) themes and subthemes to be identified from the final code structure and their relationships presented.[31]

The initial coding framework for our analysis will start from broad categories identified in the previous scoping review, on which the interviews were structured. Within these broad categories (ie, describing research gaps, experience with identifying and displaying research gaps), analytic categories will be inductively derived from the data. In this sense, our approach includes both top-down and bottom-up development of analytic categories and themes.

Trustworthiness during thematic data analysis will be ensured by storing raw data systematically, documenting detailed notes about the development and hierarchies of concepts and themes, establishing consensus on themes, providing detailed descriptions of context and describing the process of coding and analysis.[8 9] NVivo V.12 Pro, a qualitative data analysis software, will be used for data management and analysis.

## Ensuring study quality

To further ensure rigour and trustworthiness, this study will be guided by Guba and Lincoln's concepts for defining and investigating quality in qualitative research that can be considered parallel to quantitative research concepts of validity and reliability.[27 32 33] The concepts include credibility, transferability, dependability, confirmability, audit trails and reflectivity. They are inter-related, and thinking through them from the onset and incorporating them in a study will improve the study rigour.

Credibility is defined as the confidence that can be placed in the truth of the research findings[34–36]; it is considered the most important criterion to ensure rigour and trustworthiness. To ensure credibility of our study, we will use peer debriefing, which will entail the qualitative lead researcher (LN) seeking support from the senior researcher (DH) to provide scholarly guidance. The feedback will help improve the quality of the inquiry findings.[36] Transferability refers to the extent to which findings of qualitative research can be transferred to other contexts and are useful to people in other settings.[21 36–38] We aim to address transferability by reporting a rich, detailed description of the key stakeholders' context and location.[36 38] Dependability is related to whether the research questions are clear and logically connected to the research purpose and design.[37] We aim to achieve dependability by first drafting this protocol to guide our study and future studies with a similar purpose. Confirmability has been related to objectivity or neutrality for establishing that the data and interpretations of the findings are not figments of the inquirer's imagination but are clearly derived from the data, that data collection and interpretations of the study are clearly deliberated from the data and not misinterpreted.[37] We aim to address confirmability by documenting the justification of methodological and analytical choices to illustrate how the data were derived in relation to the study objectives and transparently describing the research steps taken from the start of the project to the development and reporting of the findings. Records of the research path will be kept throughout the study, and debriefing sessions will be held between the main researcher (LN) and senior researcher (DH). Finally, reflexivity includes examining one's own conceptual lens, explicit and implicit assumptions, preconceptions and values and how these affect research decisions in all phases of qualitative studies. Reflexivity will be achieved by ensuring transparency of the study process by maintaining clear documentation.

## Patient or public involvement

There is no patient or public involvement in the design or analysis of this study. However, we plan to involve patients/the public in findings that pertain to them and in disseminating study findings. This will be achieved by using patient/public online platforms such as peopleinresearch.org.

## DISCUSSION

This study will provide insights into issues related to defining research gaps and methods used to identify and display gaps in health research from perspectives of key stakeholders involved in the process. This is a follow-up study of a wider project; the first study was a scoping review exploring methods used to identify and display research gaps reported in scientific publications.[1] The scoping review showed variation and ambiguity in how research gaps are described as well as the methods used to identify and prioritise research gaps. Several of the articles described the development of a framework or tool for identifying and prioritising research gaps and applying it to a specific topic area as an example for application.[1 2 7 39] There were no evaluations of reproducibility of the method/frameworks identified in the scoping review.[1 7] Furthermore, despite articles highlighting the existence of research gaps in their studies, very few specifically described the gaps and the causes or the method of identification, so fully understanding the relevance and importance of the research gap to adequately address it is difficult. Our scoping review also primarily found the use of secondary research methods such as systematic reviews and scoping reviews as the most commonly used methods to identify gaps; although other methods were identified, they were inadequately described. The scoping review also showed that besides researchers, different audiences including clinicians, policymakers, funders and patients or the public can benefit from understanding gaps and methods/practices on how to identify and display gaps in health research. This qualitative study aims to go beyond the scientific literature in describing, identifying and displaying gaps in health research and directly talk to people about their understanding and practices. Given the nature of this topic that is not fully explored, there is a need to investigate real practices to be able to develop methodological guidance, taking into consideration the existing literature and ongoing practices.

This study has some limitations; one is not including patients/the public in designing the study. Including patients/public perspectives would have benefited the study design by being able to improve the importance and relevance of the findings for this population. One of the main strengths of the study is improving the definition of research gaps and subsequently improving the accurate reporting of research gaps to clearly elucidate the characteristics, which can help in making evidence-based decisions. For example, making a decision based on a research gap contributing to lack of primary research on a specific health problem can differ from a research gap related to lack of secondary research summarising the research. Hence, all these factors regarding research gaps need to be highlighted if they are known and made explicit when disseminating and communicating research. In addition, providing more information on what the gap represents may inform users of evidence on more specific information about the research gap and how it can be addressed more accurately. We anticipate that this study will advance efforts in research and practice on this topic area.

## ETHICS AND DISSEMINATION

Informed consent will be obtained in accordance with the University of Liverpool Ethics Committee board requirements. Verbal consent will be sought for phone interviews and written consent for in-person interviews. Confidentiality and data protection will be ensured in accordance with the University of Liverpool Ethics Committee board. All participant information will be anonymised, and hard-copy data will be stored in a locked unit. Soft-copy material will be stored in a password-protected file. On completion of the study and publication of the study results, all study material will be stored and disposed of according to the rules and regulations of the University of Liverpool. The study protocol will be stored in the data repository Zenodo.

At the end of this research project, the results will be presented at conferences and relevant meetings (eg, H2020 Project MiRoR). They will also be published in a peer-reviewed journal and as part of a doctoral thesis of the PhD fellow (LN) as well as in professional and lay magazines and presented in workshops at professional events for stakeholder groups and as online materials with good practice examples.

**Acknowledgements** We acknowledge the support of Daniela Lai and Cristian R. Montenegro in providing feedback on the interview guide. We thank Laura Smales (BioMedEditing, Toronto, ON) for editing the manuscript.

**Contributors** LN and DH conceived the study with guidance and feedback from RP and CT-S. All authors read and approved the final manuscript.

**Funding** This project is a part of a MiRoR (Methods in Research on Research)-funded PhD being undertaken by LN. MiRoR received funding from the European Union's Horizon 2020 research and innovation programme under a Marie Sklodowska-Curie grant (agreement no. 676207).

**Competing interests** None declared.

**Patient consent for publication** Not required.

**Provenance and peer review** Not commissioned; externally peer reviewed.

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
