## [Reviewer comments · BMJ Open]

ARTICLE DETAILS

TITLE (PROVISIONAL)	Key stakeholders' perspectives and experiences with defining, identifying and displaying gaps in health research: a qualitative study protocol
AUTHORS	Nyanchoka, Linda; Tudur-Smith, Catrin; Porcher, Raphaël; Hren, Darko

VERSION 1 – REVIEW

REVIEWER	Laura Esmail Patient-Centered Outcomes Research Institute, United States.
REVIEW RETURNED	17-Dec-2018

GENERAL COMMENTS	Review • The paper does not clearly make the case why this study is needed. The protocol doesn't discuss in enough detail or precision the limitations of systematic reviews to motivate the need for the study. It also does not discuss how the AHRQ review of the frameworks for determining research gaps really calls for this specific research question and study. They mention the scoping review but do not discuss it at all and yet it is the motivation for the study.• The possibly most important sentence in the background section is: "Healthcare decisions for individual patients, public health policies, and clinical guidelines should be informed by the best available research evidence while taking into consideration missing, inadequate and insufficient evidence." Yet the protocol paper does not clarify how their study will advance research/practice to address the "missing, inadequate and insufficient evidence".• Is the goal of the study a better definition of a research gap, as proposed in the opening line of the paper? Or is it a better understanding of existing or potential approaches that could be used to better characterize evidence gaps to better inform research, practice and policy decisions? In the paper, objectives of the study refers to two major activities that stakeholders will weigh in on: the identifying of research gaps; and the describing or displaying of research gaps. These are/can be different activities. Are we talking about getting input on how research gaps are displayed, such as evidence maps? Or how they are identified? Depending upon the answer, your research questions and methods could be very different.• Has a study like the one they propose ever been done before? Has a literature review been conducted searching for peer review and grey literature that looks at this or similar questions? No indication in the paper to suggest that this step has been taken and we are left to trust the authors that they've done a thorough assessment.
---

	 • Much detail is missing from the methods including rationale for the design decisions the authors made. For example:  o The proposed rationale for using qualitative methods seems to be fairly standard and is not at all applied to their study. How specifically can an exploratory qualitative study generate the data that they need to answer their specific research question as opposed to other methods? o Interviews: the authors do not state how the groups were chosen and where the categories themselves come from. Why this particular constellation of stakeholder perspectives? Insufficient rationale provided. o The interview guide lists three domains but provide little detail on how these domains were chosen. They also state that they will revise the interview guide as the study unfolds. Is this a methodologically sound approach? If so, support with evidence. o The sampling technique proposed is snowballing and convenience sampling and yet no rationale was provided to support this technique. o Sample size considerations do not seem fully thought through and on face value, largely underpowered for saturation given number of groups proposed. Furthermore, the rationale for the estimated sample size is unclear and not well-specified. • Lastly, this study missed an opportunity in not having patient and stakeholder feedback in designing the study itself. The aims of the study appear to explore different approaches to making evidence gaps more meaningful. Ironically, the proposed study suffers from exactly what the intent aims to address: lack of input from patients and stakeholders. The study may benefit from engaging them early and often, starting at the research question and design stage.
--	---

REVIEWER	Stuart Nicholls Ottawa Hospital Research Institute
REVIEW RETURNED	15-Jan-2019

GENERAL COMMENTS	Thank you for the opportunity to review the study protocol “Exploring Key Stakeholder Experiences with Defining, Identifying and Displaying Gaps in Health Research: A Qualitative Study Protocol”. I commend the authors for publishing this protocol for a qualitative study. The protocol lays out a study to explore stakeholder experiences identifying research gaps and communicating these. Given that the study design is fixed, and ethics approval has been received I limit my comments (as per the guidance provided by the journal) to the presentation of the study as opposed to making suggestions for study design, although I do offer points for consideration. GENERAL POINTS An overarching comment I would make is that the methods, at present, very much read like a text book – that is, they seem somewhat independent of the specific study, which make it unclear at times as to how certain steps specifically would occur. I would suggest that the protocol could be improved by being more specific with respect to how the methodology will be applied in the present study – especially with respect to ensuring study quality and the analytic approach (for example, more clearly articulating how the analysis will address the research questions – will you analyse the texts as a whole, or take each research question and then analyse the transcripts with this lens in mind? etc)
---

POINTS TO CONSIDER

1. The examples of organisations are largely UK based. There may be jurisdiction specific issues at play. I am sure that the authors have considered relevant factors that may affect perspectives on mapping research gaps but using these to sample (as opposed to convenience sampling) might be helpful.

2. Several questions in the interview guide (Background Q 2 and 3) may be less relevant to patients or the public if their 'work' is not their primary reason for being involved in mapping research gaps (it may be due to a health condition, for example, as opposed to their 'work' in terms of employment). Again, clarifying the adaptability of the guide and giving examples would be useful.

BACKGROUND

The NCCMT definition of 'research gap' is adopted, yet a motivation is the lack of a standard definition. Is a goal to find a consensus, to showcase variation, or to simply explore this and reasons why there is variation? Can you say more about the choice of definition here, and yet why it remains important to explore definitions rather than just adopt the NCCMT definition? Or what the implications are from varied definitions (could you give any examples?)

Indeed, I felt the background could be expanded to discuss why this is a problem – could the authors draw on the scoping review to flesh this out. Why are different approaches to mapping or identifying research gaps a problem? More exposition of the topic would be helpful.

STUDY DESIGN

There seems to be some duplication between this section and the section INTERVIEWS with respect to the justification (e.g. lines 41-46 seems to cover much of the same ground as the text in STUDY DESIGN). I would suggest that the section STUDY DESIGN could be truncated and some of the text combined with that in INTERVIEWS. Indeed, it may be worth considering a slight restructuring to rename the INTERVIEW heading to DATA COLLECTION to reflect that the interview is the data collection process.

KEY INFORMANTS

I found that I wanted more justification for the groups selected – what is it about the stakeholder types, and potentially the organisations themselves, that means that their perspective is merited. Are there theoretical reasons for their selection? Would you expect different groups to have different perspectives, in which case this might also be a reason to include them. If the authors could articulate this sort of supportive argument I think that would help clear some of the uncertainty.

Indeed, given that the study builds on a scoping review it would have been useful to see more links between the studies. For example, would there be a rationale to purposively sample participants based on the types of approaches identified in the review? Indeed, if one covers the range of approaches this might be considered a form of maximal variation sampling.

For example, one might argue that the UK NHS is a relevant organization as care is (one hopes) informed by research and so

ensuring that research meets clinical needs would be one reason why they would be a relevant body. However, you may wish to consider whether not just clinicians, but also commissioners, are relevant to the research question.

TRANSCRIPTION and THEMATIC DATA ANALYSIS

I would suggest moving the text about audio recording (page 9, lines 1-3) to precede the text about transcription given that the audio recording will precede transcription, but also logically fits with the process from audio to written word.

THEMATIC DATA ANALYSIS

Additional information would be useful in this section, and particularly applying the abstract concepts to the current study. For example, which two coders will independently code the text – will they meet and discuss? What is meant by corroboration? What constitutes corroboration in the context of qualitative coding – is it simply achieving consensus on the codes having face validity, or is it more of a line by line agreement about the coding of the text?

Can you say more about what a thematic map is? I wasn't familiar with this.

I was a little confused by the section about analytic categories. This section states that:

“We will use analytical categories to describe and explain experiences reported among the different groups of participants. The categories will be inductively derived from the data gathered by the semi-structured interviews. All data relevant to each category will be identified and examined thoroughly to develop relevant themes. This examination requires a coherent and systematic approach and involves adding categories to reflect as many of the nuances in the data as possible, rather than reducing the data. All data relevant to each category (describing research gaps, experience with identifying and displaying research gaps) will be identified and examined to ensure that each data item is checked accordingly.” (page 9)

My general understanding is that much qualitative analysis proceeds from open coding, in which coding is highly granular and quite specific, through an aggregation into broader themes. As such, it does seem to be a form of data reduction (in the sense that some nuance is invariably lost when presenting the higher-level theme). Perhaps the authors could say more about their approach and the relationship between codes, analytic categories, and themes – possibly using motivating examples?

STUDY QUALITY

I did find this section somewhat under developed. While I take the position of Guba and Lincoln in terms of offering qualitative alternatives to traditional quantitative approaches to study quality, they seemed to be superficially dealt with. I wonder if this could be pulled out and put in a table or given more detail. I was especially intrigued by the idea of transferability (which the authors relate to generalizability), given that generalizability of not a criterion considered for qualitative studies. Having the space to expand on this would be useful to clarify what is meant.

Equally, the authors state that:

	“Dependability involves participants’ evaluation of the findings, interpretation and recommendations of the study [8]. To take this into consideration, we aim to clearly outline the different steps of the project and its findings.” One step often taken is member checking – in which transcripts and/or research interpretations are fed back to the participants themselves to verify the researcher interpretations. Is this consistent with the approach taken the by team here, or are they referring more to a strategy for dissemination? If the latter, how are they establishing the participants’ evaluation of findings? Again, having project specific details of arrangements that are in place would help to clarify this PATIENT AND PUBLIC INVOLVEMENT My understanding of Patient and Public Involvement is that it generally refers to their involvement in the design or conduct of the study (i.e. as collaborators), not as participants.
--	---

REVIEWER	Anastasia Mallidou University of Victoria, BC Canada
REVIEW RETURNED	24-Feb-2019

GENERAL COMMENTS	BMJ Open – Review Article bmjopen-2018-027926 Thank you for the opportunity to review this interesting and novel topic of research. The following comments intend to improve the manuscript and clarify some unclear areas. Abstract The abstract is complete, but two areas need revision or clarification:  1. On page 1 (line 46), the proposed dissemination of the findings include “meetings”; what do the authors refer to? What kind of meeting with whom? 2. On the same page and line, dissemination includes only peer-reviewed publications; what about professional journals or magazines as well as lay magazines, since the study participants include public/patients? Strengths and limitations of the study  • This section needs revision. None of those four bullets are clearly described as strengths or limitations. • In the third bullet (p. 2, lines 18-21), the authors refer to a “follow-up study”, which is not mentioned in the abstract. Background  1. Needs more elaboration. The topic of the proposed study, although innovative, is not well described by providing background information on the issue that the authors want to address. 2. On page 3 (line 36), “the standard method for identifying research gaps” is the scoping review; not the systematic review. 3. It is unclear whether the “public” is different group of study participants than “patients”. What are their differences? Why are they mentioned separately? Purpose and Objectives □ The purpose of this study is “to explore experiences with describing, identifying, and displaying health research gaps” (p. 3, lines 44-50). This statement raises three questions:  o What kind of experiences do the authors expect the 
---

	public/patients have “with describing, identifying, and displaying health research gaps”?  ○ How patients or the public without specific knowledge on research can understand the topic of this study (i.e., research gaps), especially when there is not clear definition of “research gaps”? ○ The objectives also include “key stakeholders’ views on describing health research gaps” (p. 3, lines 53-54), which are different than/beyond experiences... Methods  1. Study design: Only the first line (p. 4, line 7) explicitly refers to the study design. The following paragraph is an instructive description of the design that I would suggest to remove. 2. Eligibility criteria (p. 4, lines 28-30): It is unclear how research users are relevant stakeholders to provide insights about research gaps. Further explanation on this would shed light to those study participants’ role.
	 Anastasia Mallidou, RN, PhD 23 February 2019 Page 1 of 2 Discussion  1. Expected outcomes (p. 8, lines 39-49): In this paragraph, the purpose of the study is described again. Also, for the first time, the authors refer to the origin of the proposed study and its connection with a previous conducted scoping review. This paragraph would be better moved to the methods section. Appendices  1. Appendix 2 (line 14): the statement refers to “the different experiences” of key stakeholders. How the authors know that the key stakeholders will have “different” experiences? 2. Appendix 3 (p. 3, lines 3-5 and 10-12): There is a discrepancy between those two paragraphs. In the first one, participants agree that their data cannot be able to withdraw after providing them. In the second paragraph (lines 10-12), participants have the right to request their already collected data to be destroyed “at any time until the data is submitted for publication”. This needs revision and further clarification. Overall, the structure of the paper is confusing. Usually, the main sections of a research paper include Background, Methods, Results, Discussion, and Conclusions.
	 Anastasia Mallidou, RN, PhD 23 February 2019 Page 2 of 2

VERSION 1 – AUTHOR RESPONSE

Author’s Response to the Reviewers

Title: Key stakeholders’ perspectives and experiences with defining, identifying and displaying gaps in health research: a qualitative study protocol

Date: 10 April 2019

Reviewer: 1

Reviewer Name: Laura Esmail

Institution and Country: Patient-Centered Outcomes Research Institute, United States.

Please state any competing interests or state 'None declared': No competing interests declared.

Please leave your comments for the authors below

See comments in attached review (Written Review - Esmail).

- The paper does not clearly make the case why this study is needed. The protocol doesn't discuss in enough detail or precision the limitations of systematic reviews to motivate the need for the study. It also does not discuss how the AHRQ review of the frameworks for determining research gaps really calls for this specific research question and study. They mention the scoping review but do not discuss it at all and yet it is the motivation for the study.

We thank the Reviewer for this important point. To address this, we have updated the protocol background section including further justification and evidence to outline the importance of this study for identifying gaps in health research and methods used to identify and display the gaps. This modification can be found on Background section Page 3(67–121).

- The possibly most important sentence in the background section is: "Healthcare decisions for individual patients, public health policies, and clinical guidelines should be informed by the best available research evidence while taking into consideration missing, inadequate and insufficient evidence." Yet the protocol paper does not clarify how their study will advance research/practice to address the "missing, inadequate and insufficient evidence".

We thank the Reviewer for this key point. We have provided further justification on the importance of this study to advance research on addressing missing, inadequate and insufficient evidence, updated on Page 4(95–110).

- Is the goal of the study a better definition of a research gap, as proposed in the opening line of the paper? Or is it a better understanding of existing or potential approaches that could be used to better characterize evidence gaps to better inform research, practice and policy decisions? In the paper, objectives of the study refer to two major activities that stakeholders will weigh in on: the identifying of research gaps; and the describing or displaying of research gaps. This are/can be different activities. Are we talking about getting input on how research gaps are displayed, such as evidence maps? Or how they are identified? Depending upon the answer, your research questions and methods could be very different.

Thank you for the comment. This study is part of a larger study aimed at improving the definitions of gaps in health research and developing methodological guidance on approaches to identifying and displaying gaps in health research. Exploring definitions of research gaps particularly based on the reason for the gap will not only inform appropriate methodological approaches to identify the gaps but also better characterize the gaps. This can better inform research, practice and policy decisions guided by more concrete and clear characteristics of the gaps in research. We have updated our objectives to address the Reviewer's suggestion. Please refer to Page 5(122–124)

- Has a study like the one they propose ever been done before? Has a literature review been conducted searching for peer review and grey literature that looks at this or similar questions? No indication in the paper to suggest that this step has been taken and we are left to trust the authors that they've done a thorough assessment.

This study is a follow-up study of an existing scoping review that searched peer-reviewed and grey literature on defining, identifying, prioritizing and displaying gaps in health research. We have now included additional information on this scoping review in relation to this qualitative study protocol. Existing studies mainly report on the use of secondary research methods to identify research gaps (systematic reviews, health technology assessments and scoping reviews). We have now updated this on Page 4(82–100).

Much detail is missing from the methods including rationale for the design decisions the authors made. For example:

- The proposed rationale for using qualitative methods seems to be fairly standard and is not at all applied to their study. How specifically can an exploratory qualitative study can

generate the data that they need to answer their specific research question as opposed to other methods?

Thank you for highlighting this point. We have now modified the protocol to reflect that this study is part of a larger project in which the first step was to conduct a scoping review and explore how to describe gaps in health research and methods used to identify and display gaps in health research in scientific literature, and the second step is to further explore the aim according to key stakeholder experiences and perspectives. We aim to gain a better understanding of what the literature reports and what key experts think and do in practice. Our main overall aim is to develop methodological guidance on defining, identifying and displaying gaps in health research. This update can be found on lines Page4–5(111–121).

- Interviews: the authors do not state how the groups were chosen and where the categories themselves come from. Why this particular constellation of stakeholder perspectives? Insufficient rationale provided.

The group selection was based on the main categories of groups identified from a scoping review conducted by the authors. Therefore, the conception and design of the study is directly linked to the scoping review, which aimed to describe methods to identify and display gaps in health research. We have further clarified this in the protocol, including more information on sampling justification and selection of categories to be interviewed, refer to Page 5–6 (134–146).

- The interview guide lists three domains but provide little detail on how these domains were chosen. They also state that they will revise the interview guide as the study unfolds. Is this a methodologically sound approach? If so, support with evidence.

Thank you for this comment. The domains were selected from a scoping review on the same topic and therefore follow the same structure of what we aimed to explore in the scoping review. We have now clarified this in the protocol. For developing the semi-structured interview guide, we also used information from the scoping review to guide the questions. The interview guide covers the main topics of the study, providing a focused structure for the discussion during the interviews but should not be followed strictly. Instead, the idea is to explore the research area by collecting similar types of information from each participant, by providing participants with guidance on what to talk about (Gill et al. 2008). We have updated this section, with further information and justification on Page 8(184–193).

- The sampling technique proposed is snowballing and convenience sampling and yet no rationale was provided to support this technique.

We thank the Reviewer for this comment. On further evaluation of our sampling methodology, we will focus on the use of purposeful sampling and justification. We selected purposeful sampling because we wanted to interview respondents with specific experiences who can give information-rich answers. This is a technique widely used in qualitative research for the most effective use of resources (eg. Patton 1990). In the recently conducted scoping review study on the same topic, we identified key persons and organizations that would be relevant to interview. In addition, when designing the scoping review study, we included expert consultations of identified key persons to contact in terms of identifying gaps in health research.

- Sample size considerations do not seem fully thought through and on face value, largely underpowered for saturation given number of groups proposed. Furthermore, the rationale for the estimated sample size is unclear and not well-specified.

In the study by Guest et al. (2006), an empirical approach was used with a set of 60 interviews, The authors concluded that saturation occurred with 12 interviews, with broader themes apparent after a mere 6 interviews, with numbers much lower than some of the suggested estimates of numbers needed that they identified in the literature. They noted that factors such as heterogeneity of the sample affected how many interviews were required but concluded that to understand common

perceptions and experiences among a group of relatively homogeneous individuals, 12 interviews should suffice. Another study by Hennink et al. (2016), on examining 25 in-depth interviews, found that code saturation was reached with 9 nine interviews, with the range of thematic issues identified. The authors proposed that 16 to 24 interviews were needed to reach meaningful saturation. Therefore, we aimed to gather 14–24 interviews distributed between the three main categories (evidence to inform future research, health practice and policy). We further clarified this on Page 6(line 152–163).

- Lastly, this study missed an opportunity in not having patient and stakeholder feedback in designing the study itself. The aims of the study appear to explore different approaches to making evidence gaps more meaningful. Ironically, the proposed study suffers from exactly what the intent aims to address: lack of input from patients and stakeholders. The study may benefit from engaging them early and often, starting at the research question and design stage.

This is a very important point and we agree with the Reviewer. Indeed, including patient feedback in the study design and conduct could improve the study, particularly to better understand how these groups make decisions when the evidence is missing, inadequate or insufficient. However, we did consider expert consultation (stakeholders) on the design of the study including developing and testing the topic guide. This has now been updated throughout the protocol, clarifying no patient/public involvement in the design or conduct of the study, specifically stated on page 10 (256) and page 11(279–281).

Reviewer: 2

Reviewer Name: Stuart Nicholls

Institution and Country: Ottawa Hospital Research Institute

Please state any competing interests or state 'None declared': None declared

Thank you for the opportunity to review the study protocol "Exploring Key Stakeholder Experiences with Defining, Identifying and Displaying Gaps in Health Research: A Qualitative Study Protocol". I commend the authors for publishing this protocol for a qualitative study.

The protocol lays out a study to explore stakeholder experiences identifying research gaps and communicating these. Given that the study design is fixed, and ethics approval has been received I limit my comments (as per the guidance provided by the journal) to the presentation of the study as opposed to making suggestions for study design, although I do offer points for consideration.

GENERAL POINTS

An overarching comment I would make is that the methods, at present, very much read like a text book—that is, they seem somewhat independent of the specific study, which make it unclear at times as to how certain steps specifically would occur. I would suggest that the protocol could be improved by being more specific with respect to how the methodology will be applied in the present study—especially with respect to ensuring study quality and the analytic approach (for example, more clearly articulating how the analysis will address the research questions—will you analyse the texts as a whole, or take each research question and then analyse the transcripts with this lens in mind? etc) We thank the Reviewer for reviewing our study protocol.

Thank you for highlighting this key point, which agrees with the other Reviewers' comments, and we have now addressed it by providing further clarification on the background of the protocol. This can be found on page 3 - 4, line 67–110. Regarding analysis, we plan to take each research question and analyze the transcripts with this in mind. We have now updated the analysis section on Page 8–9(206–216) to reflect this.

POINTS TO CONSIDER

1. The examples of organizations are largely UK based. There may be jurisdiction specific issues at play. I am sure that the authors have considered relevant factors that may affect perspectives on mapping research gaps but using these to sample (as opposed to convenience sampling) might be helpful.

Thank you for highlighting this. First, our examples are indeed primarily UK based because of both jurisdiction-specific issues but also given that the main findings from our scoping review study on the same topic mainly referenced UK-based organizations. We agree that these examples may affect perspectives on mapping research gaps and we will take this into consideration. Although it is also important to note that our interviewees are not limited to UK-based organizations or contacts and we anticipate a wider target audience than the scoping review has indicated. We have now updated our protocol to take this point into account on Page 11(281–284).

2. Several questions in the interview guide (Background Q 2 and 3) may be less relevant to patients or the public if their 'work' is not their primary reason for being involved in mapping research gaps (it may be due to a health condition, for example, as opposed to their 'work' in terms of employment). Again, clarifying the adaptability of the guide and giving examples would be useful. Thank you for your comment. We have different questions for each of the categories, which is reflected in the updated interview guide in appendix 1, specifying the follow-up questions.

BACKGROUND

The NCCMT definition of 'research gap' is adopted, yet a motivation is the lack of a standard definition. Is a goal to find a consensus, to showcase variation, or to simply explore this and reasons why there is variation? Can you say more about the choice of definition here, and yet why it remains important to explore definitions rather than just adopt the NCCMT definition? Or what the implications are from varied definitions (could you give any examples?)

Indeed, I felt the background could be expanded to discuss why this is a problem – could the authors draw on the scoping review to flesh this out. Why are different approaches to mapping or identifying research gaps a problem? More exposition of the topic would be helpful.

In this study, we adopted the definition from the National Collaborating Centre for Methods and Tools (NCCMT) in Canada, which describes a research gap as a clinical question for which missing or insufficient information limits the ability to reach a conclusion. The definition provides a general understanding of the term “research gaps” needed to explore the topic in more depth, particularly other definitions and descriptions of gaps in health research. The importance to explore definitions rather than just adopt the NCCMT definition is mainly because of the different gaps that can exist besides only missing or insufficient information that limits the ability to cover a gap, such as methodology gap, theory gap, absolute gap, among others, as identified in the scoping review findings. We have now updated this on page 3(74-81).

STUDY DESIGN

There seems to be some duplication between this section and the section INTERVIEWS with respect to the justification (e.g. lines 41-46 seems to cover much of the same ground as the text in STUDY DESIGN). I would suggest that the section STUDY DESIGN could be truncated and some of the text combined with that in INTERVIEWS. Indeed, it may be worth considering a slight restructuring to rename the INTERVIEW heading to DATA COLLECTION to reflect that the interview is the data collection process.

Thank you for this comment, we have updated it according to your suggestion and restructured the section. It now reads Data collection and recording and includes information on the interview guide (appendix1).

KEY INFORMANTS

I found that I wanted more justification for the groups selected—what is it about the stakeholder types, and potentially the organizations themselves, that means that their perspective is merited. Are there theoretical reasons for their selection? Would you expect different groups to have different perspectives, in which case this might also be a reason to include them. If the authors could articulate this sort of supportive argument I think that would help clear some of the uncertainty.

Thank you for this comment. The selection of groups is based on the scoping study research findings of organizations and groups working on identifying gaps in health research. We have updated the protocol to reflect this and provide a justification for this selection of key informants on page 5(134–146).

Indeed, given that the study builds on a scoping review it would have been useful to see more links between the studies. For example, would there be a rationale to purposively sample participants based on the types of approaches identified in the review? Indeed, if one covers the range of approaches this might be considered a form of maximal variation sampling.

Thank you for this important point, we have included more information on the scoping review study and findings in relation to this planned study. We have updated the protocol accordingly and give more information linking the studies throughout the protocol

For example, one might argue that the UK NHS is a relevant organization as care is (one hopes) informed by research and so ensuring that research meets clinical needs would be one reason why they would be a relevant body. However, you may wish to consider whether not just clinicians, but also commissioners, are relevant to the research question.

Thank you. This is a key point and we will include commissioners in our protocol. This is now updated on page 4(106).

TRANSCRIPTION and THEMATIC DATA ANALYSIS

I would suggest moving the text about audio recording (page 9, lines 1-3) to precede the text about transcription given that the audio recording will precede transcription, but also logically fits with the process from audio to written word.

Thanks for the suggestion; we have updated it accordingly in the Method and Analysis section on page 5.

THEMATIC DATA ANALYSIS

Additional information would be useful in this section, and particularly applying the abstract concepts to the current study. For example, which two coders will independently code the text—will they meet and discuss? What is meant by corroboration? What constitutes corroboration in the context of qualitative coding—is it simply achieving consensus on the codes having face validity, or is it more of a line by line agreement about the coding of the text?

We have now updated the thematic data analysis section on page 8 – 9 (206–226) with more information to highlight the two coders who will independently code the text and establish consensus. Corroboration in our protocol refers mainly to agreement between the two coders in terms on what they coded, mainly for face validity.

Can you say more about what a thematic map is? I wasn't familiar with this.

A thematic map is an illustration of the relationship between themes. We omitted this term, given that we found it unnecessary and can confuse the reader.

I was a little confused by the section about analytic categories. This section states that:

“We will use analytical categories to describe and explain experiences reported among the different groups of participants. The categories will be inductively derived from the data gathered by the semi-structured interviews. All data relevant to each category will be identified and examined thoroughly to develop relevant themes. This examination requires a coherent and systematic approach and involves adding categories to reflect as many of the nuances in the data as possible, rather than reducing the data. All data relevant to each category (describing research gaps, experience with identifying and displaying research gaps) will be identified and examined to ensure that each data item is checked accordingly.” (Page 9)

My general understanding is that much qualitative analysis proceeds from open coding, in which coding is highly granular and quite specific, through an aggregation into broader themes. As such, it does seem to be a form of data reduction (in the sense that some nuance is invariably lost when presenting the higher-level theme). Perhaps the authors could say more about their approach and the relationship between codes, analytic categories, and themes—possibly using motivating examples? Thank you for this comment. We aim to develop a coding framework based on the broad categories identified in the scoping review and then apply an inductive (granular, bottom-up) approach within those broad categories. In that sense, our approach is both top-down and bottom-up. We have further clarified this on page 9(210–221).

STUDY QUALITY

I did find this section somewhat under developed. While I take the position of Guba and Lincoln in terms of offering qualitative alternatives to traditional quantitative approaches to study quality, they seemed to be superficially dealt with. I wonder if this could be pulled out and put in a table or given more detail. I was especially intrigued by the idea of transferability (which the authors relate to generalizability), given that generalizability of not a criterion considered for qualitative studies. Having the space to expand on this would be useful to clarify what is meant.

Equally, the authors state that:

“Dependability involves participants' evaluation of the findings, interpretation and recommendations of the study [8]. To take this into consideration, we aim to clearly outline the different steps of the project and its findings.”

One step often taken is member checking—in which transcripts and/or research interpretations are fed back to the participants themselves to verify the researcher interpretations. Is this consistent with the approach taken the by team here, or are they referring more to a strategy for dissemination? If the

latter, how are they establishing the participants' evaluation of findings? Again, having project specific details of arrangements that are in place would help to clarify this

We thank the Reviewer for highlighting these points and fully agree with them. We have now elaborated further in this section including dependability according to Tobin and Begley, 2004 as they clearly relate it to study reliability and whether the research questions are clear and logically connected to the research purpose and design. We have updated the study protocol to provide more information on page 10 (239–242).

PATIENT AND PUBLIC INVOLVEMENT

My understanding of Patient and Public Involvement is that it generally refers to their involvement in the design or conduct of the study (i.e. as collaborators), not as participants.

Thank you for this point. We do agree that this project does not involve patients and therefore updated this to reflect this versus mentioning how patients or the public participate in the study. We have also addressed this as one of the limitations of our study given that patient or public involvement would have enriched the study design and relevance for this group. This is updated on page 11(276–279).

Reviewer #3: Anastasia Mallidou, RN, PhD

Thank you for the opportunity to review this interesting and novel topic of research. The following comments intend to improve the manuscript and clarify some unclear areas.

Abstract

The abstract is complete, but two areas need revision or clarification:

1. On page 1 (line 46), the proposed dissemination of the findings include “meetings”; what do the authors refer to? What kind of meeting with whom?

We thank the Reviewer for highlighting this. We have further elaborated on this to specify the type of meeting on page 2 (53)

2. On the same page and line, dissemination includes only peer-reviewed publications; what about professional journals or magazines as well as lay magazines, since the study participants include public/patients?

We thank the Reviewer for the suggestions. We have now included this in the abstract and protocol, also on page 2 (54–55)

Strengths and limitations of the study

- This section needs revision. None of those four bullets are clearly described as strengths or limitations.

We thank the Reviewer and have taken the suggestion to account. We have now revised this section to align better with the study strengths and limitations on page 2(57 -65).

- In the third bullet (p. 2, lines 18-21), the authors refer to a “follow-up study”, which is not mentioned in the abstract.

Thank you for highlighting this point. We have now revised the abstract to provide more information on what we mean by the study protocol is a follow-up study, i.e., a scoping review on page 1 (48 -51).

Background

1. Needs more elaboration. The topic of the proposed study, although innovative, is not well described by providing background information on the issue that the authors want to address.

We have added further justification on the importance and existing literature on the topic area in the background section, highlighting the issue of aiming to further explore the terms used to refer to gaps in health research and methods used to identify and display gaps in health research. This area has very little research in the topic area, particularly on specific methods used when identifying gaps in health research. We have now updated the background section with supporting evidence on previous studies in relation to the topic area on page 3–4.

2. On page 3 (line 36), “the standard method for identifying research gaps” is the scoping review; not the systematic review.

Thank you for this clarification, indeed more recent evidence shows that scoping reviews are the standard methods for identifying gaps in health research. We have updated this in the background section with references to reflect the literature on both systematic reviews and scoping reviews for identifying gaps in health research. This is updated on page 3(80–92).

3. It is unclear whether the “public” is different group of study participants than “patients”. What are their differences? Why are they mentioned separately?

We agree with the Reviewer that this is rather confusing and have combined them to read as public or patients.

Purpose and Objectives

- The purpose of this study is “to explore experiences with describing, identifying, and displaying health research gaps” (p. 3, lines 44-50). This statement raises three questions:
 - What kind of experiences do the authors expect the public/patients have “with describing, identifying, and displaying health research gaps”?

We are interested in how the public/patients use evidence particularly when the information in the evidence is missing or insufficient to make treatment choices or health decisions. We have now

updated our interview guide demonstrating the different questions in the supplementary material (appendix 1).

- How patients or the public without specific knowledge on research can understand the topic of this study (i.e., research gaps), especially when there is not clear definition of “research gaps”?

This is a very important point, and indeed part of the purpose to explore. We do think this could be a difficult area to explore and could be an area for future research, particularly given that we had no patient/public involvement in the design of the study, yet relied on the scoping review findings on the same topic. Therefore, a future project focusing primarily on this audience will be of key importance to inform this area further.

- The objectives also include “key stakeholders’ views on describing health research gaps” (p. 3, lines 53-54), which are different than/beyond experiences...

Thank you for this point. We are interested in both perspectives and experiences on the topic area. We have now updated the protocol accordingly.

Methods

1. Study design: Only the first line (p. 4, line 7) explicitly refers to the study design. The following paragraph is an instructive description of the design that I would suggest to remove.

We agree with the Reviewer. We have taken this into account and removed it.

2. Eligibility criteria (p. 4, lines 28-30): It is unclear how research users are relevant stakeholders to provide insights about research gaps. Further explanation on this would shed light to those study participants’ role.

Thank you for this point. We have added additional justification on how the research users are relevant stakeholders to provide insights about research gaps under the section (Study sample and recruitment). This is now updated on page 5 (123–153).

Discussion 1.

Expected outcomes (p. 8, lines 39-49): In this paragraph, the purpose of the study is described again. Also, for the first time, the authors refer to the origin of the proposed study and its connection with a previous conducted scoping review. This paragraph would be better moved to the methods section. Thanks for this suggestion. We have now moved this paragraph to the methods section of the protocol, and also titled the expected outcomes section as discussion.

Appendices

1. Appendix 2 (line 14): the statement refers to “the different experiences” of key stakeholders. How the authors know that the key stakeholders will have “different” experiences?

Thank you for this comment. We have updated the protocol accordingly and removed “the different experiences” of key stakeholders in Appendix 2; indeed, this is unknown if the key stakeholder will have “different” experiences. We have now updated it accordingly.

3. Appendix 3 (p. 3, lines 3-5 and 10-12): There is a discrepancy between those two paragraphs. In the first one, participants agree that their data cannot be able to withdraw after providing them. In the second paragraph (lines 10-12), participants have the right to request their already collected data to be destroyed “at any time until the data is submitted for publication”. This needs revision and further clarification.

We have now revised this, providing further clarification, by mainly focusing on voluntary participation of the study and omitting the following sentence that was unclear.

Overall, the structure of the paper is confusing. Usually, the main sections of a research paper include Background, Methods, Results, Discussion, and Conclusions.

Thank you for your suggestion. We have modified the structure of the protocol and it now includes Background, Methods and Analysis, Discussion and Dissemination as the main headings.

VERSION 2 – REVIEW

REVIEWER	Laura Esmail Patient-Centered Outcomes Research Institute, United States
REVIEW RETURNED	23-Apr-2019

GENERAL COMMENTS	- While the study appears to be important, the paper remains somewhat confusing to read, which may make it inaccessible to the readers. There are also typos and comments left in the margin from previous drafts. The paper may benefit from having an editor or another senior scientist revise the manuscript for structure, style, clarity and grammar. - It might be worthwhile for the authors to also link the work to the effort to reduce research waste (i.e., Chalmers and Glasziou). Doing so would link it to a larger effort to increase the value and quality of research worldwide. Just a suggestion. - I thank the authors for clarifying the degree of patient/public/stakeholder involvement in the design of the study. It is not too late to incorporate some level of patient/stakeholder involvement in the conduct, analysis and/or dissemination of the study findings. Just a suggestion but it may increase the relevance of the work and uptake of study findings.
--

REVIEWER	Stuart Nicholls Ottawa Hospital Research Institute
REVIEW RETURNED	26-Apr-2019

GENERAL COMMENTS	The reviewer also provided a marked copy with additional comments. Please contact the publisher for full details.
---

REVIEWER	Anastasia Mallidou University of Victoria, Canada
REVIEW RETURNED	22-Apr-2019

GENERAL COMMENTS	The manuscript is very well written. I have only one comment in "Strengths & Limitations" paragraph: It is unclear whether the first and third bullet are strengths or limitations; I would suggest more elaboration on those points. Good work!
--

VERSION 2 – AUTHOR RESPONSE

Reviewer: 1

Reviewer Name: Laura Esmail

Institution and Country: Patient-Centered Outcomes Research Institute, United States.

Please state any competing interests or state 'None declared': No competing interests declared.

- While the study appears to be important, the paper remains somewhat confusing to read, which may make it inaccessible to the readers. There are also typos and comments left in the margin from previous drafts. The paper may benefit from having an editor or another senior scientist revise the manuscript for structure, style and clarity and grammar.

We thank the Reviewer for the second review and important comments to improve the readability and quality of the protocol. We have taken into account the comments and revised the protocol accordingly, with assistance from a senior scientist and editor.

- It might be worthwhile for the authors to also link the work to the effort to reduce research waste (i.e., Chalmers and Glasziou). Doing so would link it to a larger effort to increase the value and quality of research worldwide. Just a suggestion.

We indeed agree on the importance of linking this study with the ongoing efforts on reducing research waste, we have now updated the protocol on page 4(104-106).

- I thank the authors for clarifying the degree of patient/public/stakeholder involvement in the design of the study. It is not too late to incorporate some level of patient/stakeholder involvement in the conduct, analysis and /or dissemination of the findings. Just a suggestion that it may increase the relevance of the work and update of study findings.

We thank the Reviewer for this recommendation on patient/stakeholder involvement. As stated on the protocol, we included stakeholder involvement in the design of the study, particularly experts in the field to help clarify key terms and concepts that were used in the conceptualization of the scoping review, which informed the design of this qualitative study. In terms of patient involvement, taking into account your recommendation we aim to disseminate our study findings to patient/public groups. We have now updated this on page 6 (145 – 147) and page 10(256- 258).

Reviewer: 2

Reviewer Name: Stuart Nicholls

Institution and Country: Ottawa Hospital Research Institute

Please state any competing interests or state 'None declared': None declared

- We thank the Reviewer for his comments and suggestion to improve the study protocol. We have now addressed each of the items suggested and updated the protocol by highlighting each item in yellow. Please see amendments on the revised manuscript with highlighted revisions.

Reviewer #3: Anastasia Mallidou, RN, PhD

Thank you for the opportunity to review this interesting and novel topic of research. The following comments intend to improve the manuscript and clarify some unclear areas.

The manuscript is very well written. I have only one comment in “Strengths & Limitations” paragraph: It is unclear whether the first and third bullet are strengths or limitations; I would suggest more elaboration on those points. Good work!

We thank the Reviewer for reviewing our manuscript and providing additional valuable feedback to improve the study. We have now updated the “strengths and limitations” paragraph to clarify which are our strengths and limitations. This is updated on page 2(58-63).

VERSION 3 – REVIEW

REVIEWER	Laura Esmail Patient-Centered Outcomes Research Institute, USA
REVIEW RETURNED	10-Jul-2019

GENERAL COMMENTS	I thank the authors for addressing the outstanding comments.
--

REVIEWER	Stuart Nicholls Ottawa Hospital Research Institute
REVIEW RETURNED	19-Jul-2019

GENERAL COMMENTS	Good luck
-----------